# First Detection of SARS-CoV-2 in White Rhinoceros during a Small-Scale Coronavirus Surveillance in the Bandia Reserve, Senegal

**DOI:** 10.3390/ani13162593

**Published:** 2023-08-11

**Authors:** Jignesh Italiya, Vojtěch Vacek, Petr Matějů, Christophe Dering, Seyma S. Celina, Arame Ndiaye, Jiří Černý

**Affiliations:** 1Center for Infectious Animal Diseases, Faculty of Tropical AgriSciences, Czech University of Life Sciences Prague, 165 00 Prague, Czech Republicjiricerny@ftz.czu.cz (J.Č.); 2Department of Zoology and Fisheries, Faculty of Agrobiology, Food, and Natural Resources, Czech University of Life Sciences Prague, 165 00 Prague, Czech Republic; vacekvojtech@af.czu.cz; 3Department of Animal Science and Food Processing, Faculty of Tropical AgriSciences, Czech University of Life Sciences Prague, 165 00 Prague, Czech Republic; matejup@ftz.czu.cz; 4Reserve De Bandia, Sindia 23000, Senegal; chrdering@gmail.com; 5Centre d’Études pour la Génétique et la Conservation (CEGEC S.A.S.U.), Dakar 10455, Senegal; emarasn@gmail.com

**Keywords:** SARS CoV-2, coronaviruses, wildlife surveillance, molecular detection

## Abstract

**Simple Summary:**

This study focuses on the molecular surveillance of coronaviruses in wildlife in Senegal. Fecal samples were collected from various species of wild animals, both in the Bandia Reserve and Ngaparou. The results revealed the absence of coronaviruses in hedgehogs, non-human primates, and a giraffe. However, a positive sample obtained from a white rhinoceros yielded SARS-CoV-2 through sequencing of the RdRp gene. This finding represents the first documented case of molecular detection of SARS-CoV-2 in white rhinoceros, expanding our understanding of potential hosts of the virus. This finding expands our understanding of potential hosts of SARS-CoV-2 and highlights the importance of using wildlife monitoring to improve coronavirus surveillance.

**Abstract:**

The SARS-CoV-2 pandemic has heightened interest in the monitoring and surveillance of coronaviruses in wildlife. Testing for the virus in animals can provide valuable insights into viral reservoirs, transmission, and pathogenesis. In this study, we present the results of the molecular surveillance project focused on coronaviruses in Senegalese wildlife. During the project, we screened fecal samples of the wild animals living in the Bandia Reserve (ten non-human primates, one giraffe, and two white rhinoceros) and the free-living urban population of African four-toed hedgehogs in Ngaparou. The results showed the absence of coronaviruses in hedgehogs, non-human primates, and a giraffe. A single positive sample was obtained from a white rhinoceros. The sequencing results of amplified RdRp gene confirmed that the detected virus was SARS-CoV-2. This study represents the first documented instance of molecular detection of SARS-CoV-2 in white rhinoceros and, therefore, extends our knowledge of possible SARS-CoV-2 hosts.

## 1. Introduction

Coronaviruses (CoVs) infect a wide range of animal species, having a particular affinity for their respiratory and intestinal systems. The severity of infections caused by these viruses can vary greatly, ranging from asymptomatic cases to fatal outcomes [1]. Coronaviruses (*Coronaviridae* family, *Orthocoronavirinae* subfamily) have a single-stranded positive-sense RNA genome (+ssRNA), which is the largest among all known viruses, spanning a length of 25 to 33 kilobases [2]. The genomic RNA of CoVs is capable of functioning as an mRNA and is considered infectious in its purified form [1]. The subfamily *Orthocoronavirinae* is divided into four genera (*Alphacoronavirus*, *Betacoronavirus*, *Gammacoronavirus*, and *Deltacoronavirus*), and SARS-CoV-2 belong to the genus *betacoronavirus*.

While bats (Chiroptera) have long been recognized as the primary reservoir of CoVs, various other mammalian species also harbor specific CoVs. Hedgehogs, in particular, exhibit a high susceptibility to CoVs, as evidenced by documented occurrences of MERS-CoV-related CoVs in several European countries, including France, Germany, Italy, the United Kingdom, and Poland [3,4,5]. Furthermore, research has identified the possibility of coronavirus infections in non-human primates. For instance, wild chimpanzees in Côte d’Ivoire [6] and native hamadryas baboons (*Papio hamadryas hamadryas*) in Saudi Arabia have been found to be susceptible to these viruses [7].

In Africa, surveillance efforts to detect coronavirus nucleic acid in non-bat wildlife, livestock, and domestic animals have been limited. Several surveillance studies have been conducted, including the investigation of human coronavirus OC43 (HCoV-OC43) transmission between humans and chimpanzees in Côte d’Ivoire, MERS-CoV-specific monitoring in livestock animals in Ghana, and general surveillance among wild animals in Gabon [8]. The overall proportion of positive coronavirus RNA detected in these studies was less than 1%. The findings include positive cases identified in non-human primate (14 chimpanzees), ungulate (1 bush duiker), carnivore (1 African palm civet), and rodent species (13 individuals) [8].

Over the past two decades, the *Coronaviridae* family has been associated with three significant epidemic and pandemic outbreaks, which were primarily attributed to the *betacoronavirus* genus. The most recent and notable of these outbreaks was the COVID-19 in humans caused by SARS-CoV-2, which has shed light on the potential for reverse zoonosis, wherein viruses can transmit from humans to animals. Over the past three years, several instances of symptomatic and asymptomatic natural SARS-CoV-2 infection have been reported in various animal species, further emphasizing the potential for interspecies transmission [9,10]. Molecular and serological diagnostic methodologies are frequently used to detect SARS-CoV-2 infections in animal and human populations. Various laboratory techniques are commonly employed to characterize strains implicated in outbreaks, such as RT-PCR, RT-LAMP, virus isolation, and sequencing, including next-generation sequencing [11]. Several cases of active SARS-CoV-2 infections have been identified in zoos, free-ranging wild animals, and domestic animals through pathogen-specific surveillance studies [9]. There have been documented instances of lions, tigers, cougars, leopards, lynx, otters, coati, giant anteater, binturong, and gorillas being confirmed to be positive for SARS-CoV-2 using PCR and genetic-sequencing techniques [9,12,13,14,15,16]. The virus neutralization test (VNT), the surrogate virus neutralization test (sVNT), and the enzyme-linked immunosorbent assay (ELISA) have been used to detect previous exposure to SARS-CoV-2 by evaluating antibody immune responses. In order to detect antibodies against nucleoprotein (N), a commercial double-antigen polyspecific ELISA has been used for all susceptible animal species, demonstrating high sensitivity [17]. For instance, in July 2021, a study revealed that 40% of free-ranging white-tailed deer (*Odocoileus virginianus*) tested positive for antibodies against SARS-CoV-2 in four US states. This finding identified white-tailed deer as a wildlife host of the disease, providing evidence of their susceptibility to SARS-CoV-2 infection [18]. Additionally, a case of natural SARS-CoV-2 infection was recorded in a free-range black-tailed marmoset (*Mico melanurus*) studied in an urban area in the Central-West region of Brazil, highlighting the occurrence of the virus in non-human primate populations [19]. The study conducted in the Campania region of Italy revealed the existence of serological evidence indicating SARS-CoV-2 infection in lactating cows. However, the investigation did not detect the existence of neutralizing antibodies against bovine coronavirus (BCoV) [20].

These examples highlight the significance of understanding the potential role of various animal species in the transmission and maintenance of CoVs, including SARS-CoV-2. Monitoring and surveillance efforts across diverse wildlife populations are vital to the identification of potential reservoirs, the assessment of the risk of zoonotic transmission, and the implementation of appropriate preventive measures to mitigate future outbreaks.

This study aimed to investigate the potential presence of CoVs in two distinct populations: the fauna of the Bandia Reserve and the free-living four-toed hedgehogs (*Atelerix albiventris*) in Ngaparou town. The rationale behind this investigation stems from our university’s ongoing long-term monitoring project, which focuses on studying the wildlife inhabiting these specific areas.

## 2. Materials and Methods

### 2.1. Sample Collection and Study Region

This study focused on two specific sites in western Senegal: the Bandia Reserve and the coastal town of Ngaparou (Figure 1).

The Bandia Reserve, which spans an area of 3500 hectares and is located 65 km from Dakar, was established in 1990 with the aim of conserving wildlife. The reserve boasts a diverse ecosystem, harboring more than 120 bird species and 18 large animal species, both native and non-native to Senegal. Ngaparou, on the other hand, is a coastal town situated 75 km south of Dakar and located 33 km away from the Bandia Reserve [21].

In May 2022, fresh animal fecal samples were collected from both the Bandia Reserve and Ngaparou. Twenty hedgehogs were captured in Ngaparou in order to obtain fecal samples. Subsequently, the captured animals were released back into their original habitat. In the Bandia Reserve, ten samples were collected randomly from non-human primates inhabiting the area. The reserve is home to two distinct species of non-human primates: patas monkeys (*Erythrocebus patas*) and green vervet monkeys (*Chlorocebus pygerythrus*), which coexist within the same habitat. Due to the similarities in the dimensions and morphology of their fecal matter, it was challenging to differentiate the source species or individual based on the collected samples. Additionally, observations indicated that these two species often reside in the same social groups, further complicating the identification process.

Furthermore, three fresh fecal specimens were obtained from the Bandia Reserve, consisting of two samples from white rhinoceros (*Ceratotherium simum*) and one sample from a giraffe (*Giraffa camelopardalis*). To ensure sample integrity, all collected samples were promptly stored on ice and processed on the day of collection.

### 2.2. Nucleic Acid Extraction and RT-PCR

The field-based RNA extraction process was conducted using the Quick-DNA/RNA Viral MagBead kit (Zymo Research, Irvine, CA, USA). This kit utilizes magnetic bead-based techniques that do not require centrifugation, enabling the extraction of RNA from freshly collected fecal samples in the field. A total of 33 samples (20 samples from four-toed hedgehogs, 10 samples from non-human primates, 2 samples from white rhinoceros, and 1 sample from a giraffe) were subjected to RNA analysis using RT-PCR. The one-step RT-PCR kit (QIAGEN, Germantown, MD, USA) and the portable miniPCR^®^ mini8 thermal cycler (miniPCR, Cambridge, MA, USA) were employed for this purpose.

The RT-PCR system consisted of a 25-microliter reaction volume containing the following components: 5 μL of 5× QIAGEN OneStep RT-PCR buffer, 1 μL of dNTP (resulting in a final concentration of 400 μM for each dNTP), 1 μL each of upstream and downstream primers (at a concentration of 25 μmol/L), 0.25 μL of RNAsin (at a concentration of 40 μ/μL), 1 μL of enzyme mix, and 2 μL of RNA template; the remaining volume was filled with RNase-free water to reach a total volume of 25 μL.

Two distinct sets of primers were used to selectively amplify specific regions within the RNA-dependent RNA polymerase (RdRP) gene, which is a highly conserved gene among coronaviruses. The first primer set consisted of forward (5′-AARTTYTAYGGHGGYTGG-3′) and reverse (5′-GARCARAATTCATGHGGDCC-3′) primers targeting a 668-base pair fragment of the polymerase gene. The experiment commenced via an initial reverse transcription process at a temperature of 50 °C for a period of 30 min. This step was followed by PCR activation at 95 °C for 15 min. The amplification phase consisted of 35 cycles, each involving 40 s at 94 °C, 40 s at 52 °C, and 1 min at 72 °C. Finally, a final extension step was carried out at 72 °C for 10 min, as described by Hu et al. [22]. Similarly, the second primer set consisted of forward (5′-GGGDTGGGAYTAYCCHAARTGYGA-3′) and reverse (5′-TARCAVACAACISYRTCRTCA-3′) primers targeting a 452-base pair fragment of the polymerase gene. The experiment commenced via an initial reverse transcription process at a temperature of 50 °C for a period of 30 min. This step was followed by PCR activation at 95 °C for 15 min. The amplification phase consisted of 35 cycles, each involving 40 s at 94 °C, 40 s at 50 °C, and 1 min at 72 °C. Finally, a final extension step was carried out at 72 °C for 10 min, as described by Hasoksuz M et al. [23]. The RT-PCR products were visualized using a portable electrophoresis system BlueGel™ (miniPCR, USA).

To ensure the accuracy of the results, the RT-PCR screening conducted in the field did not incorporate a positive control to mitigate the risk of false positives resulting from cross-contamination. However, the effectiveness of the RT-PCR reactions in producing positive results using positive control samples was separately evaluated in a laboratory setting at the standard university laboratory. This evaluation was carried out before the commencement of the in-field experiment, ensuring the reliability of the in-field RT-PCR screening process.

### 2.3. Cloning and Sequencing of RT-PCR Products

The amplified products of RT-PCR positive samples were sent to the Center for Infectious Animal Diseases (FTZ) in Prague for sequencing in order to circumvent any potential legal complications associated with the transfer of biological specimens. Prior to sequencing, these transported amplified products underwent a cloning process in the pJET vector to enhance the quality of the resulting sequences. The amplification product was treated to create blunt ends and then ligated into the pJET1.2/blunt vector. Subsequently, Sanger sequencing was performed on the resulting plasmid using two plasmid-specific primers provided by the pJET2.1 vector: the forward sequencing primer (5′-d(CGACTCACTATAGGGAGAGCGGC)-3′) and the reverse sequencing primer (5′-d(AAGAACATCGATTTTCCATGGCAG)-3′). The obtained sequence data were analyzed using the Geneious software (Version 2022.2) and compared to existing sequences in the GenBankTM dataset via basic local alignment tool (BLAST) analysis [24].

## 3. Results

Out of the 33 fecal samples collected from both the Bandia Reserve and Ngaparou city, 32 samples tested negative for coronaviral RNA. However, a notable finding emerged when one sample from a white rhinoceros yielded a positive result for coronavirus RNA when using the second primer set, resulting in an amplicon of 452 bp in length. No clinical signs or symptoms suggestive of any infection were observed in the rhinoceros within the Bandia Reserve throughout the study period.

To gain further insights into the identified coronavirus, a pair-end Sanger sequencing analysis was conducted, and subsequent Blast analysis revealed that the sequence belonged to the SARS-CoV-2. Due to the relatively short length of the fragment and its high level of conservation, further genetic characterization of the sequence was not feasible.

## 4. Discussion

SARS-CoV-2, which is a coronavirus initially identified in Wuhan, China, in late 2019, has rapidly spread worldwide, leading to the COVID-19 pandemic [25]. Since the emergence of COVID-19 pandemic, numerous cases of SARS-CoV-2 infection in animals have been reported [26]. Observational and experimental studies on a range of non-human mammalian species, including free-living, captive, domestic, and farmed animals, have identified at least 54 species susceptible to the virus [27].

In order to monitor the potential presence of SARS-CoV-2 in animals, extensive surveillance programs have been implemented. For instance, a study conducted from January to March 2021 focused on monitoring free-ranging white-tailed deer in Northeast Ohio, revealing their vulnerability to COVID-19 through real-time RT-PCR testing [14]. Furthermore, in India, SARS-CoV-2 was detected in a free-ranging leopard (*Panthera pardus fusca*) [28], and cases of natural infection in captive wild animals in zoos are well documented [29]. These instances emphasize the need for comprehensive risk analysis to evaluate the potential transmission of the virus from animals to humans. Additionally, continuous surveillance is crucial to gain a deeper understanding of the role that animals play in the spread of the virus.

In this study, we identified a positive case of coronavirus infection in a white rhinoceros using the RT-PCR assay. Subsequent sequencing of a short fragment of the RdRp gene confirmed the presence of SARS-CoV-2 in the rhinoceros’ sample. Comparisons between the rhinoceros host cell entry receptor ACE2 and its human counterpart ACE2 revealed homology, suggesting the potential for SARS-CoV-2 infection in rhinoceros [30]. However, it is important to note that in silico studies solely focusing on host cell entry may have limitations, as successful viral replication could also rely on various other factors, such as tissue proteases TMPRSS2, CTSL, or ADAM-17 [31]. Further investigations are required to fully understand the susceptibility and implications of SARS-CoV-2 infection in rhinoceros and its potential role in the transmission dynamics of the virus.

The results identified in the Bandia Reserve, which is a semi-enclosed wildlife sanctuary known for tourism, raise concerns about the potential transmission of infections to animals. This concern is primarily due to the possibility of direct or indirect human interaction with wild animals through activities such as providing feed, which is a common practice, or engaging in wildlife safari tours. It is important to consider that the fresh fecal sample collected from the white rhinoceros may also be influenced by environmental contaminants.

Currently, our understanding of SARS-CoV-2 biology in rhinoceroses is limited, underscoring the need to continue surveillance studies in rhinoceros populations to identify any similar occurrences and potential spillover events. In our investigation, we found no evidence of coronaviruses in the four-toed hedgehog from Ngaparou, as well as in non-human primates and giraffes from the Bandia Reserve. It is worth noting that throughout the period of sample collection in May 2022, the prevalence of SARS-CoV-2 in the human population was minimal, with most instances ranging from zero to a maximum of twelve cases [32].

This study, to our best knowledge, represents the first documented instance of molecular detection of SARS-CoV-2 in white rhinoceros, but it does have a few limitations. Firstly, it is important to note that the investigation was carried out on a limited number of samples, thereby limiting the generalizability of the findings to the entire populations of four-toed hedgehogs, patas and green vervet monkeys, and giraffes. Consequently, it is not possible to definitively conclude that these animal populations were entirely free of the CoVs. The primary objective of the study was to assess the prevalence of the CoVs in animals; therefore, the use of specific primers targeting SARS-CoV-2 during the fieldwork was not prioritized. Instead, the focus was on detecting the presence of coronaviruses or bovine-like coronaviruses in general. In such cases, sequencing of the RT-PCR amplicon alone would have been sufficient to identify and differentiate the viral presence. Furthermore, the field conditions presented logistical challenges, as we lacked deep freezers for long-term storage of virus RNA. Consequently, we opted to transport the more stable DNA amplicon for sequencing, as opposed to the relatively unstable virus RNA. Subsequently, to circumvent any potential legal complications associated with transporting biological samples, only the PCR products were transported to the Czech Republic for sequencing. This decision resulted in our inability to perform amplification of the spike (S) or receptor-binding domain (RBD) genes for the purpose of identifying variants. Another limitation of this study is that we cannot definitively rule out the possibility of the passive transit of the virus through the digestive system. Our sampling methodology aimed to minimize the likelihood of detecting virus remnants from passive transit. By directly collecting fresh fecal samples from the animals in their natural habitats, we aimed to capture active shedding of the virus, which would indicate an active infection rather than passive transit. Future research should address these limitations to gain a comprehensive understanding of the status of coronaviruses in the studied animal populations.

## 5. Conclusions

In conclusion, this study sheds light on the presence of coronaviruses in wildlife populations in Senegal, specifically in the Bandia Reserve and Ngaparou. While no coronaviruses were detected in four-toed hedgehogs, non-human primates, and a giraffe, the molecular surveillance revealed the presence of SARS-CoV-2 in a white rhinoceros. This finding expands our understanding of potential hosts of SARS-CoV-2 and highlights the importance of wildlife monitoring for coronavirus surveillance. To obtain a comprehensive understanding of the prevalence, transmission, and impact of coronaviruses in wildlife populations, as well as to elucidate the dynamics of viral spillover events, further research and enhanced surveillance measures are warranted.

## Figures and Tables

**Figure 1 animals-13-02593-f001:**
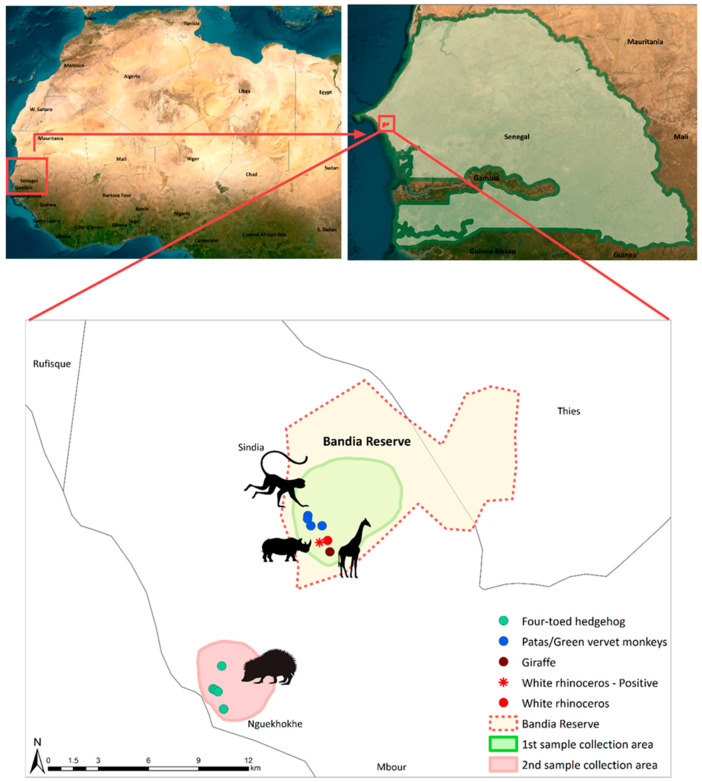
Map of the location and sampling points of the study area. The sampling locations of the four-toed hedgehog, patas and green vervet monkeys, giraffe, and rhinoceros are depicted using green, blue, dark red, and red circles, respectively. The red asterisk highlights the positive sample derived from a white rhinoceros. The green background depicts the sampling site inside of the Bandia Reserve, where fresh fecal samples of patas and green vervet monkeys, giraffe, and white rhinoceros were collected. The light red area shows the locations in Ngaparou at which fresh fecal samples of the four-toed hedgehog were collected. The boundaries of the Bandia Reserve are represented using a red dotted line. The figure was generated using ArcMap 10.8.2.

## Data Availability

The FASTA files obtained via Sanger dideoxy sequencing were deposited in the GenBank nucleotide database (https://www.ncbi.nlm.nih.gov/nuccore/OR262347; accessed on 17 July 2023) under accession number: OR262347.

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
