# Peer review of "First Detection of SARS-CoV-2 in White Rhinoceros during a Small-Scale Coronavirus Surveillance in the Bandia Reserve, Senegal"

_animals, 2023, doi:10.3390/ani13162593_

Round 1

Reviewer 1 Report

The study “First Detection of SARS-CoV-2 in White Rhinoceros During Coronavirus Surveillance in the Bandia Reservoir and Senegal” by Italiya reporting the detection of SARS_COV2 in rhinoceros. However the claim is not strongly supported as the authors only sequenced a highly conserved region and did only a simple blast search without showing the results. Other comments include:

1.       Introduction: please expand your introduction on the light of recent publications, e.g.

https://doi.org/10.51585/gjm.2021.2.0007

https://doi.org/10.1126/science.abb7015

2.       for Sample collection do the authors confirm that each sample represents a different animal to count them as different samples

3.        

4.       line 129: correct to “in the field”

5.       line 166: please correct to “using the second primer”

6.       lines 166-168: how the authors mointored the rhinoceros within the Bandia Reserve throughout the study and how the fecal sample was ensured it belongs to the same animal?

7.       Lines 170-172: the authors just did a blast analysis and based on it the authors claimed it is SARS-COV-2, the high level of conservation may suggest it is just a coronavirus, not necessirly SARS-COV2, please justify or show your analysis

8.       Line 214: cases (reference?).?

none

Author Response

Dear Editor,

Please find attached our revised version of the manuscript entitled "First Detection of SARS-CoV-2 in White Rhinoceros During Small-Scale Coronavirus Surveillance in the Bandia Reserve, Senegal". We hope that the revised version now addresses all reviewers’ comments and suggestions.

We would like to thank the reviewers for contributing their time in reading our work and providing feedback and comments. We have outlined below the responses to their comments and where we have implemented their suggestions.

Reviewer #1

> The study “First Detection of SARS-CoV-2 in White Rhinoceros During Coronavirus Surveillance in the Bandia Reservoir and Senegal” by Italiya reporting the detection of SARS COV2 in rhinoceros. However the claim is not strongly supported as the authors only sequenced a highly conserved region and did only a simple blast search without showing the results. Other comments include: Introduction: please expand your introduction on the light of recent publications, e.g. https://doi.org/10.51585/gjm.2021.2.0007; https://doi.org/10.1126/science.abb7015.

We amended the introduction to include the suggested study - https://doi.org/10.1126/science.abb7015. Lines: 70
Second reference - https://doi.org/10.51585/gjm.2021.2.0007 was not utilized in the introduction, as it does not align with the topic of the manuscript, which focuses on "COVID-19: Risk assessment and mitigation measures in healthcare and non-healthcare workplaces". However, Introduction improved using other latest publications. Lines: 74-93

> For sample collection do the authors confirm that each sample represents a different animal to count them as different samples?

In our study, each sample represents a distinct animal. To ensure this, we conducted observations of the animals in the Bandia reserve in collaboration with animal rangers. Subsequently, we collected fresh faecal samples from each individual animal, confirming that each sample corresponds to a distinct individual. We highlighted this in Methods section. For the sample collection from hedgehogs, Animals were captured to collect faecal samples before releasing back to the original habitat. Lines: 113-123

> Line 129: correct to “in the field”; Line 166: please correct to “using the second primer”; Line 214: cases (reference?)?
             DONE

>  Lines 166-168: how the authors monitored the rhinoceros within the Bandia Reserve throughout the study and how the fecal sample was ensured it belongs to the same animal?

To obtain fresh faecal samples from the animals inhabiting in the Bandia Reserve, we employed two approaches. Firstly, in the reserve, we conducted observations alongside experienced animal rangers to monitor and identify individual animals, thereby ensuring that each collected sample represented a unique contributor. Samples of rhinoceros were collected using the same method, and the reserve has only two rhinoceros, making animal identification simple.

Secondly, with a specific focus on Hedgehog faecal samples, we took a slightly different approach. We rounded up all the hedgehogs from the field and subsequently collected their samples after releasing them back into the same location. This approach allowed us to collect the faeces of hedgehogs and differentiate each individual by ensuring which faecal sample belongs to which individual. We clearly highlighted this point in Method section. Lines: 113-127

> Lines 170-172: the authors just did a blast analysis and based on it the authors claimed it is SARS-COV-2, the high level of conservation may suggest it is just a coronavirus, not necessirly SARS-COV2, please justify or show your analysis.

Thanks to the reviewer for this comment. We need to highlight that although the target region exhibits high conservation within individual viruses, we observed a notable interspecific difference that facilitated virus identification. In our BLAST search - with default settings but a maximum number of target sequences set to 5000 -, we found a total of 5017 hits, all of which were identified as SARS-CoV-2. It is important to mention that the BLAST search did not reveal any other closely related betacoronaviruses, including SARS-CoV-1. As a result, we can reasonably conclude that the virus detected in the rhino faeces corresponds to SARS-CoV-2. Further, we submitted our sequence data to GenBank with Accession ID: OR262347

Reviewer 2 Report

The manuscript titled "a" by Jignesh et al. describes the first recording of Sras Cov-2 in the white rhinoceros. Although the work is quite well written and has a good scientific impact, some methodological weaknesses and the absence of data included in this version of the work preclude publication. Here are some of my general comments:

Title: Is it appropriate to speak of "Coronavirus Surveillance" for a few tested wild animals? This concept has also been stated in the Simple Summary, line 22.

Introduction: I suggest to the authors that they better define the state of the art regarding the detection of SARS-CoV-2 in animals (both domestic and wild) as well as the description of exposure to this pathogen (such as dogs, cats, pangolins, mink, cattle, other ruminants, etc.). In this regard, the authors could also draw a table containing the signaling of the pathogen or the detection of antibodies in the various domestic species (since the manuscript lacks tables and the bibliographic references could be improved). This table could also be inserted into discussions.

Materials and methods:

The main problem with this manuscript lies in this section. The authors themselves describe the protocol used as amplifying a highly conserved sequence among coronaviruses (thus common among various coronaviruses). The second set of primers, which should be highly specific for SARS-CoV-2, is composed of degenerative primers, amplifying regions other than those commonly used to characterize SARS-CoV-2. The bibliographic references to these protocols are missing, or, in any case, the authors should better describe and justify the choice made. Even the application used is questionable: with field kits, without DNA extraction, without positive control. This approach is feasible in the field but needs to be accompanied by more robust laboratory analyses.

Results:

Line 163: Throughout the manuscript, the authors emphasize the freshness of the collected feces. I think it's not important.

Line 171: "Due to the relatively short length of the fragment and its high level of conservation, further genetic characterization of the sequence was not feasible." I don't think that's the real reason why full sequencing and characterization haven't been done.

Data relating to the sequencing of the amplified sequence is missing (possibly as supplementary data). Are you sure this sequence is exclusive to SARS-CoV-2 and not common to the various beta-coronaviruses?

Discussion:

Line 214: "reference"?

Line 216: The limitations of this study have not been accurately listed.

Line 228: At the end of the manuscript, the authors list the main genomic markers used for the detection and characterization of SARS-CoV-2 in humans and animals. Why weren't they used? Authors must provide explanations.

Even if it were actually SARS-CoV-2 and not other beta-coronaviruses (which should in any case be excluded), the authors never took into consideration the possible passive transit that the virus could have carried out in the digestive system of the animal (to later be found by the authors).

English is ok.

Author Response

Dear Editor,

Please find attached our revised version of the manuscript entitled "First Detection of SARS-CoV-2 in White Rhinoceros During Small-Scale Coronavirus Surveillance in the Bandia Reserve, Senegal". We hope that the revised version now addresses all reviewers’ comments and suggestions.

We would like to thank the reviewers for contributing their time in reading our work and providing feedback and comments. We have outlined below the responses to their comments and where we have implemented their suggestions.

Reviewer #2

The manuscript titled "a" by Jignesh et al. describes the first recording of Sras Cov-2 in the white rhinoceros. Although the work is quite well written and has a good scientific impact, some methodological weaknesses and the absence of data included in this version of the work preclude publication. Here are some of my general comments:

> Title: Is it appropriate to speak of "Coronavirus Surveillance" for a few tested wild animals? This concept has also been stated in the Simple Summary, line 22.

Thanks to the reviewer for his/her comment. "Coronavirus Surveillance" might be overstated for a study involving only a few tested wild animals. The term "surveillance" typically implies systematic and widespread monitoring efforts involving a large population or area. To address this concern, we have revised the title to better align with the main objective of the study.

> Introduction: I suggest to the authors that they better define the state of the art regarding the detection of SARS-CoV-2 in animals (both domestic and wild) as well as the description of exposure to this pathogen (such as dogs, cats, pangolins, mink, cattle, other ruminants, etc.). In this regard, the authors could also draw a table containing the signalling of the pathogen or the detection of antibodies in the various domestic species (since the manuscript lacks tables and the bibliographic references could be improved). This table could also be inserted into discussions.

We amended the introduction based on your suggestions. Please see lines: 66-93

> Materials and methods: The main problem with this manuscript lies in this section. The authors themselves describe the protocol used as amplifying a highly conserved sequence among coronaviruses (thus common among various coronaviruses). The second set of primers, which should be highly specific for SARS-CoV-2, is composed of degenerative primers, amplifying regions other than those commonly used to characterize SARS-CoV-2. The bibliographic references to these protocols are missing, or, in any case, the authors should better describe and justify the choice made. Even the application used is questionable: with field kits, without DNA extraction, without positive control. This approach is feasible in the field but needs to be accompanied by more robust laboratory analyses.

We amended the text to include your comments in Methods section. Please see lines: 147-152, 157-161, 162-169. In this study, the field kit used eliminates the need for DNA extraction, making the process more efficient to employ in the field. Regarding the choice of primers, our rationale was driven by the expectation of detecting other betacoronaviruses, such as bovine or bovine-like coronaviruses. In such cases, sequencing of the RT-PCR amplicon would provide adequate information, rendering further steps unnecessary.

> Results: Line 163: Throughout the manuscript, the authors emphasize the freshness of the collected feces. I think it's not important.

We amended the text based on your suggestion.

> Line 171: "Due to the relatively short length of the fragment and its high level of conservation, further genetic characterization of the sequence was not feasible." I don't think that's the real reason why full sequencing and characterization haven't been done.

Regrettably, the primary reason for not conducting whole genome sequencing was due to constraints we faced. For the analyses, we utilized RT-PCR directly in the field with the help of a mobile PCR device (miniPCR). While conducting field sequencing of positive samples (e.g., using minION) was not initially planned, there were valid justifications for this decision. Our expectation was to detect other betacoronaviruses, such as bovine or bovine-like coronaviruses. In such scenarios, sequencing of the RT-PCR amplicon would suffice to provide the necessary insights. Additionally, the lack of deep freezers in the field presented challenges for long-term storage of virus RNA. Consequently, we could only transport the stable DNA amplicon for sequencing, as the relatively unstable virus RNA was not feasible for preservation during transportation.

> Data relating to the sequencing of the amplified sequence is missing (possibly as supplementary data). Are you sure this sequence is exclusive to SARS-CoV-2 and not common to the various beta-coronaviruses?

Our detected SARS-CoV 2 sequence data was submitted to GenBank with Accession ID: OR262347

> Discussion: Line 214: "reference"?

DONE

> Line 216: The limitations of this study have not been accurately listed.

Dear Reviewer please see lines: 245-268 where we listed the limitations of our study in the end of Discussion part. Firstly, it is important to acknowledge that our study was conducted with a limited number of samples, which may impact the generalizability of the findings to a broader population. This is also the reason why we changed our title from “Coronavirus Surveillance” to “Small-Scale Coronavirus Surveillance”. Secondly, we made a deliberate decision not to use specific SARS-CoV-2 primers, as it was in line with the specific aim of our study. While this choice had its reasoning, it might have affected the sensitivity and specificity of the results in detecting the virus. Lastly, we faced potential legal complications related to transporting biological samples, which led us to opt for a different approach. Consequently, only the PCR products were transported to the Czech Republic for sequencing. This decision, unfortunately, resulted in the inability to perform amplification of the spike (S) or receptor-binding domain (RBD) genes, which are crucial for identifying variants of the virus.

> Line 228: At the end of the manuscript, the authors list the main genomic markers used for the detection and characterization of SARS-CoV-2 in humans and animals. Why weren't they used? Authors must provide explanations.

The reason behind this was our expectation of potentially detecting other betacoronaviruses, such as bovine or bovine-like coronaviruses. In such cases, sequencing of the RT-PCR amplicon alone would have been sufficient to identify and differentiate the viral presence. Furthermore, the field conditions presented logistical challenges, as we lacked deep freezers for long-term storage of virus RNA. Consequently, we opted to transport the more stable DNA amplicon for sequencing, as opposed to the relatively unstable virus RNA. We hope that this explanation clarifies the reasoning behind our methodology. Please see lines: 250-260

> Even if it were actually SARS-CoV-2 and not other beta-coronaviruses (which should in any case be excluded), the authors never took into consideration the possible passive transit that the virus could have carried out in the digestive system of the animal (to later be found by the authors).

We appreciate the reviewer's concern regarding the potential for passive transit of SARS-CoV-2 or other beta-coronaviruses in the digestive system of the animal. While our research detected SARS-CoV-2 in white rhinoceros, we acknowledge the possibility of passive transit of the virus through the digestive system. Although we cannot definitively rule out this possibility, our sampling methodology aimed to minimize the likelihood of detecting virus remnants from passive transit. By collecting fresh faecal samples directly from the animals in their natural habitat, we aimed to capture active shedding of the virus, which would indicate an active infection rather than passive transit. However, we highlighted this in the last part of Discussion where we mentioned the limitations of our study. Please see lines: 260-265

Round 2

Reviewer 1 Report

none

Author Response

No comments or suggestions were found. However, minor changes are being carried out according to suggestions from Reviewer 2. 

> The paper has been revised and included in the newly submitted version based on recommendations. Please see lines: 20, 34, 66, 93-96.

Reviewer 2 Report

I am glad to see that the authors have addressed the points of my previous review, providing additional information to the manuscript and depositing the obtained sequence in Genbank. The work carried out remains modest, but carried out with methodology. Furthermore, the limitations of the study were described and discussed. I believe the manuscript is ready for acceptance. I recommend the authors to delete “to our best knowledge” (line 20 and 34) and I suggest a reference that can further enhance the introduction (Line 90). 10.3390/ani12111459

Author Response

Dear Editor,

Please find attached our revised version of the manuscript entitled "First Detection of SARS-CoV-2 in White Rhinoceros During Small-Scale Coronavirus Surveillance in the Bandia Reserve, Senegal". We hope that the revised version now addresses all reviewers’ comments and suggestions.

We would like to thank again the reviewers for contributing their time in reading our work and providing feedback and comments. We have outlined below the responses to their comments and where we have implemented their suggestions.

Reviewer #2

  • I am glad to see that the authors have addressed the points of my previous review, providing additional information to the manuscript and depositing the obtained sequence in Genbank. The work carried out remains modest, but carried out with methodology. Furthermore, the limitations of the study were described and discussed. I believe the manuscript is ready for acceptance. I recommend the authors to delete “to our best knowledge” (line 20 and 34) and I suggest a reference that can further enhance the introduction (Line 90). 10.3390/ani12111459.

The paper has been revised and included in the newly submitted version based on recommendations. Lines: 20, 34, 66, 93-96.